# Responses of Yield and Photosynthetic Characteristics of Rice to Climate Resources under Different Crop Rotation Patterns and Planting Methods

**DOI:** 10.3390/plants13040526

**Published:** 2024-02-15

**Authors:** Hong Yang, Guangyi Chen, Ziyu Li, Wei Li, Yao Zhang, Congmei Li, Mingming Hu, Xingmei He, Qiuqiu Zhang, Conghua Zhu, Fahong Qing, Xianyu Wei, Tian Li, Xuyi Li, Yuyuan Ouyang

**Affiliations:** 1Crop Research Institute, Sichuan Academy of Agricultural Sciences, Chengdu 610066, China; yanghon222@163.com (H.Y.); chenguangyi@stu.sicau.edu.cn (G.C.); lwrice@163.com (W.L.); zchsicau@163.com (C.Z.); 2College of Agronomy, Sichuan Agricultural University, Chengdu 611130, China; lee18382498813@163.com (Z.L.); m13154632628@163.com (Y.Z.); licongmei2024@163.com (C.L.); 2020201029@stu.sicau.edu.cn (M.H.); 2020301136@stu.sicau.edu.cn (X.H.); qqZHANG299@163.com (Q.Z.); lit@sicau.edu.cn (T.L.); 3Environmentally Friendly Crop Germplasm Innovation and Genetic Improvement Key Laboratory of Sichuan Province, Chengdu 610066, China; 4Agriculture and Rural Bureau of Mianzhu, Deyang 618200, China; qingfahong@163.com (F.Q.); h2023111002@163.com (X.W.)

**Keywords:** rice, crop rotation pattern, planting method, climate factor, yield

## Abstract

Climate is the most important environmental factor influencing yield during rice growth and development. To investigate the relationships between climate and yield under different crop rotation patterns and planting methods, three typical rotation patterns (vegetable–rice (V), rape–rice (R), and wheat–rice (W)) and two mechanical planting methods (mechanical transplanting (T1) and mechanical direct seeding (T2)) were established. The results showed that compared to the V rotation pattern, the average daily temperature (ADT) during the sowing to heading stage increased under both R and W rotation patterns, which significantly shortened the growth period. Thus, the effective accumulated temperature (EAT), photosynthetic capacity, effective panicle (EP), and spikelet per panicle (SP) under R and W rotation patterns significantly decreased, leading to reductions in grain yield (GY). VT2 had a higher ratio of productive tillers (RPT), relative chlorophyll content (SPAD), leaf area index (LAI), and net photosynthetic rate (Pn) than those of VT1, which significantly increased panicle dry matter accumulation (DMA), resulting in an increase in GY. Although RT2 and WT2 had a higher RPT than those of RT1 and WT1, the GY of RT1 and WT1 decreased due to the significant reductions in EAT and photosynthetic capacity. Principal component analysis (PCA) showed that the comprehensive score for different rotation patterns followed the order of V > R > T with VT2 ranking first. The structural equation model (SEM) showed that EAT and ADT were the most important climate factors affecting yield, with total effects of 0.520 and −0.446, respectively. In conclusion, mechanical direct seeding under vegetable–rice rotation pattern and mechanical transplanting under rape–rice or wheat–rice rotation pattern were the rice-planting methods that optimized the climate resources in southwest China.

## 1. Introduction

The formation of rice yield is influenced by cultivars, cultivation methods, environment, and other factors [1,2]. Among them, climate is the most active and variable environmental factor, exerting an irreversible impact on the growth and development of rice [3,4]. Previous studies have found that the key climate factor affecting rice yield is the effective accumulated temperature (EAT). The daily active accumulated temperature, daily effective accumulated temperature, and their utilization rates during the growth period are significantly positively correlated with the spikelet per panicle (SP), 1000-grain weight (GW), seed-setting rate (SS), and grain yield (GY). Furthermore, the increase in temperature will accelerate the growth process of rice, leading to a shorter growth period and a lower utilization rate of temperature and light [5,6].

The yield depends on the accumulation and distribution of photosynthetic substances, with temperature exerting a significant effect on the photosynthetic capacity of rice [7,8]. Studies have shown that a high temperature can reduce the leaf area index (LAI), chlorophyll content, and net photosynthetic rate (Pn), resulting in a decrease in dry matter accumulation (DMA) [9]. Conversely, a low temperature will slow decrease the growth rate of rice, affect rice photosynthesis, and reduce total dry matter production [10,11].

Developing light, simple, and efficient cultivation techniques based on ensuring yield is the main trend of rice production in China. At present, the primary mechanized planting methods employed in China are mechanical transplanting (T1) and mechanical direct seeding (T2) [12]. Rice growth periods differ under different planting methods, which result in a different temperature and light resource environment, affect the growth process and yield formation of rice [13,14]. It is reported that compared to the transplanting method, the direct seeding method significantly increases the average daily temperature (ADT) during the sowing to heading stage, accelerates rice growth and development, shortens the growth period, and significantly reduces the effective accumulated temperature [15,16]. Changes in the growth period and effective accumulated temperature further affect the photosynthetic matter production of rice [17,18].

Southwest China is a representative region for multi-cropping grain production in the country, characterized by the rotational cultivation of rice with vegetable, rape, and wheat [19]. Due to the harvest date of previous crops, the sowing date of rice varies under different rotation patterns, resulting in great changes in temperature and light conditions during the rice growing season. These variations exert a pronounced impact on the growth process and yield formation of rice [20]. Selecting appropriate planting methods according to different crop rotation patterns can not only effectively promote the rice production potential but also improve the utilization of temperature and light resources [21]. Previous studies have proved that there had been significant differences in temperature and light resource of rice under different rotation patterns or planting methods. However, most research studies are limited to a single rotation pattern or planting method. There is limited research that investigates light and simplified cultivation techniques under multiple rotation patterns and planting methods.

Therefore, a field experiment was conducted to study the characteristics of rice growth, development, and yield formation, and their relationships with climate factors under three rotation patterns (vegetable–rice, rape–rice, and wheat–rice) and two planting methods (mechanical transplanting and mechanical direct seeding). The objectives of this study were to determine the differences among multiple crop rotation patterns and planting methods and to provide theoretical and practical basis for light and simple cultivation and efficient utilization of climate resources in multi-cropping grain production areas.

## 2. Results

### 2.1. Rice Growth Process

Different crop rotation patterns and planting methods had significant differences during the sowing to heading, heading to maturity, and whole growth periods (Table 1). The whole growth period in 2022 was 2.3 days (d) shorter than that observed in 2021, which was mainly due to the continuous high temperature during sowing to heading stage that accelerated the growth process, thus shortening the growth period by an average of 2.5 d. No significant difference was found at the heading to maturity stage. The variation in the rice growth period was consistent between the two years. The whole growth period between different crop rotation patterns followed the trend of V > R > W, with growth periods of 144.5–145.5 d, 130.5–133.5 d, and 120–123 d, respectively. The whole growth period of pattern V was 12–14 d longer than that of pattern R and 22.5–24.5 d longer than that of pattern W, whereas the R pattern was 10.5 d longer than the W pattern. Under the same rotation pattern, the whole growth period of T2 was significantly shorter than that of T1. The whole growth periods of V, R, and W patterns under T2 were shortened by 21–25 d, 23–25 d, and 18–18 d, respectively, when compared to T1. In conclusion, with the delay in the sowing date, the rice growth period was significantly shortened. Mechanical transplanting could increase the whole growth period of rice by extending the nutrient growth period when compared to mechanical direct seedling.

### 2.2. Effective Accumulated Temperature, Average Daily Temperature, and Rain Fall

There were significant differences in EAT, ADT, and rain fall (RF) among different crop rotation patterns, planting methods, years, and their interactive effects (Table 2). In 2021, compared to the V pattern, the EAT, ADT, and RF during the sowing to heading stage increased by 0.41%, 6.89%, and −1.08%, respectively, under the R pattern, and by −0.52%, 15.03%, and 1.24%, respectively, under the W pattern. During the heading to maturity stage, the EAT, ADT, and RF decreased by 11.33%, 5.38%, and −2.51%, respectively, under the R pattern, and by 30.10%, 11.15%, and −42.04%, respectively, under the W pattern. During the whole growth period, the EAT decreased by 2.74% under the R pattern and by 7.62% under the W pattern, while the ADT and RF increased by 4.11% and 0.48%, respectively, under the R pattern, and by 7.71% and 18.85%, respectively, under the W pattern. Similar trends were found in 2022, except for an increase of 3.07% in the EAT and a decrease of 101.46% in the RF under the W pattern during the sowing to heading stage in comparison to those under the V pattern. The two-year results showed that under the same crop rotation pattern, both the EAT and RF followed the order of T1 > T2 during the whole growth period, while the ADT showed an opposite trend. In summary, the V pattern had a higher EAT and ADT after heading, contributing to an enhancement in the formation of yield. Both the R and W patterns increased the ADT, thereby accelerating the growth process of rice.

### 2.3. Leaf Area Index, Relative Chlorophyll Content, and Net Photosynthesis

During the grain filling process, the LAI, SPAD, and Pn all first increased and then decreased, and peaked at 7 d after heading (Figure 1). Among different rotation patterns, the maximum values of LAI, SPAD, and Pn were obtained under the V pattern and followed the trend of V > R > W. At 7 d after heading, the LAI, SPAD, and Pn under the V pattern were 3.98–4.86%, 0.99–5.96%, and 0.92–3.95% higher than those under the R pattern, and were 8.38–11.65%, 1.25–6.42%, and 1.34–3.78% higher than those under the W pattern, respectively. Among the different planting methods, the LAI, SPAD, and Pn under VT2 significantly increased by 4.34–8.98%, 3.29–4.19%, and 3.03–7.40%, respectively, when compared to those under VT1. Additionally, the LAI, SPAD, and Pn under RT1 and WT1 significantly increased by 5.69–5.80%, 3.29–3.91%, and 2.41–2.48%, and by 4.77–8.04%, 1.65–4.46%, and 1.41–2.33%, respectively, when compared to those under RT2 and WT2. In conclusion, rice plants demonstrated superior photosynthetic characteristics under VT2, RT1, and WT1, which was conducive to photosynthesis and matter accumulation.

### 2.4. Tillering Dynamics

The tillering number (TN) and RPT were mainly affected by the crop rotation pattern, planting method, and their interaction (Table 3). During the development of growth process, the TN of each treatment showed a gradually decreasing trend. At the jointing stage, no significant difference in the TN was found between the V and W patterns, but they were both significantly higher than that of the R pattern. Compared to the R pattern, the TN increased by 4.22–4.39% under the V pattern and by 3.67–3.84% under the W pattern. Under the same crop rotation pattern, the TN of T1 was significantly higher than that of T2. Among them, the TN of VT1, RT1, and WT1 increased by 22.58–23.24%, 30.14–30.40%, and 34.34–34.63%, respectively, when compared to those of VT2, RT2, and WT2. At the heading stage, the TN under the V pattern was significantly higher than that under the R and W patterns, showing increases of 9.97–10.68% and 9.22–11.03%, respectively. No significant difference was found between VT1 and VT2, while the TNs of RT1 and WT1 were significantly higher than those of RT2 and WT2, exhibiting increases of 11.85–14.54% and 10.14–11.80%, respectively. At the maturity stage, the TN under different crop rotation patterns followed the order of V > R > W, with significant differences. Among them, the average TN under the V pattern was 6.59–6.74% higher than the R pattern and 14.22–15.62% higher than the W pattern. The RPT under different treatments ranged from 59.84% to 87.74%. The RPTs of VT2, RT2, and WT2 were significantly higher by 20.41–20.99, 9.36–10.14, and 14.75–15.84 percentage points than those of VT1, RT1, and WT1, respectively. In conclusion, compared to VT1, VT2 exhibited a higher EP and RPT. RT2 and WT2 had a higher RPT but lower effective panicles due to the decreased tillering number at the jointing stage compared to RT1 and WT1.

### 2.5. Dry Matter Accumulation

Different crop rotation patterns, planting methods, and their interactions had significant effects on DMA (Figure 2). At heading stage, the DMA under different crop rotation patterns was mainly distributed in the stem sheath, following the order of R > V > W with a significant difference. Among them, the DMA under the R pattern was 2.95–2.96% higher than the V pattern and 4.91–7.31% higher than the W pattern. The total DMA under the R pattern was 0.75–2.36% higher than the V pattern and 6.16–6.24% higher than the W pattern. Compared to VT1, the DMA in the stem sheath of VT2 significantly increased by 6.50–14.41%.

During the heading to maturity stage, the dry matter increases of HS-MS and panicle under different crop rotation patterns followed the order of V > W > R. Among them, the dry matter increases of HS-MS and panicle under the V pattern were 56.45–74.91% and 19.43–28.28% higher than those under the R pattern, and were 36.10–44.37% and 17.83–25.89% higher than those under the W pattern, respectively. Compared to T1, the dry matter increases of HS-MS and panicle under the V pattern significantly increased by 2.39–15.35% and 4.48–15.65%, respectively. Conversely, these indicators significantly decreased by 34.68–52.86% and 8.08–22.98% under the R pattern, and by 52.36–148.04% and 31.57–35.29% under the W pattern, respectively. With the development of rice growth process, nutrients in rice plants were gradually transported to the panicle, resulting in a panicle DMA of 6.95–9.79 t ha^−1^ at the maturity stage. The highest DMA was observed under the V pattern, which significantly increased by 12.15–16.31% in comparison to the R pattern and by 11.59–18.39% in comparison to the W pattern. Among the different planting methods, the panicle DMA under VT2 significantly increased by 3.68–12.01% when compared to that under VT1. Additionally, the panicle DMA significantly increased by 6.47–12.80% under RT1 and 24.50–28.49% under WT1, when compared to those under RT2 and WT2. In conclusion, rice plants could obtain a higher dry matter increase of HS-MS and a higher dry matter increase of the panicle under VT2, RT1, and WT1, which was conducive to high yield.

### 2.6. Yield and Yield Components

Crop rotation patterns, planting methods, and their interactions had significant effects on rice yield and yield components (Table 4). Both GY and EP under different crop rotation patterns tended to be in the order of V > R > W. The GY and EP under the V pattern were 10.21–22.00% and 6.87–12.63% higher than those under the R pattern, and were 13.99–24.49% and 7.14–13.03% higher than those under the W pattern, respectively. Among the different planting methods, the GY under VT2 increased by 0.90–3.33% and the EP under VT2 increased by 5.88–6.70%, when compared to that under VT1. Conversely, compared to that under RT1, the GY under RT2 significantly decreased by 8.19–8.33%, and the EP under RT2 significantly decreased by 7.75–13.36%. Similarly, the GY under WT2 significantly decreased by 4.56–6.03% and the EP under WT2 significantly decreased by 1.58–3.16%, when compared to those under WT1. Under the same crop rotation pattern, the SP of T1 was significantly higher than that of T2, and the two-year average increases in VT1, RT1, and WT1 were 4.96–5.38%, 7.24–12.19%, and 3.09–4.77%, respectively, when compared to those of VT2, RT2, and WT2. No obvious trend was found between the SS and GW. In conclusion, rice plants under the vegetable–rice rotation pattern exhibited a higher population spikelet number than that under other rotation patterns, among which VT1 demonstrated a higher spikelet per panicle than VT2, while VT2 had a higher effective panicle than VT1.

### 2.7. Principal Component Analysis

A total of 25 important indexes were used to do the correlation analysis (Figure 3). According to the correlation analysis result, the PCA of 10 indexes including climate, photosynthetic, and yield components under different treatments was used to establish a comprehensive evaluation model (Figure 4). The results showed that the cumulative contribution rate of PC1 and PC2 was 85.00%, and 8.50 variables were explained. Among them, the absolute value of the eigenvector of PC1 in order from high to low was EAT, SP, and EP, while the absolute value of the eigenvector of PC2 in order from high to low was Pn, RF, and SPAD. The comprehensive evaluation results showed that the V pattern obtained the highest comprehensive score while the W pattern obtained the lowest score (Table 5). The VT1 had the highest score in PC1, but the lowest score in PC2. Thus, the comprehensive score of VT1 was lower than that of VT2. Both RT1 and WT1 had higher scores in PC1 when compared to those of RT2 and WT2, respectively, as well as the comprehensive score. In conclusion, these results indicated that VT2, RT1, and WT1 exhibited a higher EAT, SP, EP and photosynthetic capacity, thereby effectively enhancing rice yield.

### 2.8. Structural Equation Model

After the normalization of all the data and based on qualitative analysis results, EAT, ADT, whole growth period day, EP, SP, and SS were selected as the factors affecting rice yield. SPSS AMOS 28 software was used to establish the SEM and the results showed that the selected indexes explained 82.00% of the yield variation (Figure 5). Path analysis showed that the EP, SS, and whole growth period day played leading roles in affecting yield with the direct effects of 0.70, 0.35, and 0.55, respectively, and the EP, SS, and growth period were directly affected by EAT and ADT. Among them, EAT had a positive effect on EP with a direct effect of 0.65 and whole growth period day with a direct effect of 0.71 while exerting a negative effect on SS with a direct effect of −0.61. ADT had a positive effect on SS with a direct effect of 0.45, and had a negative effect on EP with a direct effect of −0.64 and whole growth period day with a direct effect of −0.65. In conclusion, these results indicated that grain yield under different crop rotation patterns and planting methods differences were mainly caused by the variations in the EAT and ADT. These variations subsequently resulted in differences in the EP, SS, and growth period.

## 3. Discussion

### 3.1. Responses of Effective Accumulated Temperature, Daily Average Temperature, and Rain Fall to Crop Rotation Patterns and Planting Methods

The formation of rice yield depends on environmental resources such as temperature, light, and rain fall [22]. Among these, temperature is the most important factor affecting rice growth and development. Either too high or too low of a temperature is not conducive to rice growth and filling [23,24]. Previous studies have shown that the EAT during the growth period of rice is significantly positively correlated with SP, SS, GW, and GY. Additionally, ADT has been identified as an important factor influencing the growth process of rice [25]. In this study, we found that there were significant differences in the growth process, EAT, and ADT of rice corresponding to the delayed harvest dates of vegetable, rape, and wheat. It is known that, with the delay of transplanting or sowing date, the EP, SP, and GY tend to decrease [26], which is due to the shortening of the whole growth period and the decrease in the EAT of rice [15,27]. Compared to that under the V pattern, the whole growth period of rice was shortened by 12–14 d under the R pattern and 22.5–24.5 d under the W pattern. Consequently, the EAT decreased by 2.74–7.62% under the R pattern and by 0.73–3.92% under the W pattern, resulting in decreases in yield of 10.21–22.00% and 13.99–24.49%, respectively, which is in agreement with previous studies [27]. Moreover, delayed transplanting and sowing dates correlated with an increasing trend in ADT, which is consistent with earlier studies [28].

Studies have shown that different planting methods have significant effects on the temperature and light utilization of rice [29]. Conventionally, the mechanical transplanting method is associated with a higher whole growth period EAT compared to the mechanical direct seeding method [30]. However, in this study, the whole growth period EAT of VT2 showed no significant difference in relation to VT1 in 2022 but decreased by 3.63% in 2021. However, the two-year results demonstrated a higher GY with the mechanical direct seedling method than the mechanical transplanting method. This might be attributed to significant increases in SPAD, LAI, and Pn after heading using the mechanical direct seedling method. On the contrary, the EAT, SPAD, LAI, and Pn at each growth stage in both RT2 and WT2 were lower than those of RT1 and WT1, resulting in a significant decrease in GY.

Taken together, these results indicated that different crop rotation patterns combined with suitable planting methods could promote the utilization efficiency of temperature and light resources and improve the photosynthetic production capacity of rice, thus obtaining high yield. The EAT, LAI, SPAD, and Pn under VT2, RT1, and WT1 were significantly higher than those under VT1, RT2, and WT2.

### 3.2. Effects of Different Crop Rotation Patterns and Planting Methods on Rice Yield

Climate resources such as temperature, light, and rain fall are the main factors affecting rice yield [31]. Appropriate cultivation methods can effectively improve the production efficiency of temperature and light, ultimately leading to increased yield [32,33,34]. In this study, significant differences in the EAT, ADT, RF, growth period process, and yield of rice at each growth stage were found under different crop rotation patterns and planting methods. Correlation analysis results showed that GY was significantly positively correlated with the whole growing period at each growth stage, EAT after the heading stage, whole growing period EAT, ADT after the heading stage, RF during the sowing to heading stage, tillering at each growth stage, above-ground dry matter accumulation at the maturity stage, and each photosynthetic index. Conversely, GY exhibited significant negative correlations with ADT during the sowing to heading stage, whole growth period ADT, and rain fall during the heading to maturity stage.

GY consists of many evaluation indexes, and the most important factor of these that affects the formation of GY cannot be determined only from correlation analysis results. PCA is one of the most widespread statistical methods that convert multiple indexes into multiple comprehensive indexes and determine their relative importance, and ensure that the amount of information after a dimensionality reduction is maintained at a sufficiently high level [35,36]. SEM is a method for establishing, estimating, and testing causal relationship models, and has been applied to reveal and test hypothetical models, as well as to discover the interactions between variables [37]. PCA results showed that PC1 mainly included EAT, spikelet per panicle, and effective panicles, while PC2 mainly included Pn, RF, and SPAD, with RF exerting a negative impact. The top three score treatments in PC1 were VT1, VT2, and RT1, indicating superior performance in EAT, spikelet per panicle, and effective panicles. In PC2, the top three score treatments were VT2, WT2, and WT1, indicating enhanced performance in Pn, RF, and SPAD. Overall, the top three comprehensive score treatments were VT2, VT1, and RT1, with VT2 exhibiting the highest yield. SEM results showed that EAT and ADT played dominant roles in rice yield with total effects of 0.520 and −0.446, respectively. EAT positively influenced rice yield, while ADT exhibited a negative effect. Path analysis showed that the direct effects of the EAT and ADT on rice yield were limited, and their primary influence occurred indirectly by affecting EP, SS, and growth period, with indirect effects of 0.792 and −0.545, respectively.

## 4. Materials and Methods

### 4.1. Experimental Site

This experiment was conducted in 2021–2022 at the experimental base of the Grain-Economic Complex Expert Compound in Mianzhu, Sichuan Province (N 31°15′, E 104°13′). The experiment site comprised loam soil, and the previous crops consisted of vegetables (Chuanqing 2009), rape crops (Chuanyou 58), and wheat (Chuanmai 104). The rice cultivar was R534 (indica rice), and the basic physical and chemical properties of soil were determined with a reference to the soil agrochemical analysis protocol [38]. The basic physical and chemical properties of soil in each crop rotation mode are shown in Table 6.

### 4.2. Experimental Design

A two-factor randomized block design was adopted in this experiment. There were 3 crop rotation patterns (vegetable–rice rotation, rape–rice rotation, and wheat–rice rotation) and 2 planting methods (mechanical transplanting and mechanical direct sowing). Each treatment was performed with three repetitions, and the plot area was 10 m ∗ 13 m. The transplanted rice was raised for 30 d prior to transplanting to ensure that the transplanting date was the same as the direct sowing date [15]. The transplanting period of vegetable–rice, rape–rice, and wheat–rice rotation (the sowing period of mechanical direct seeding) was April 15, May 5, and May 25, respectively.

The blanket seedling technique was adopted for mechanical transplanting, and the wet seeding technique was adopted for mechanical direct seeding. For both methods, the plants were sown into 30 cm × 20 cm holes (row spacing × plant spacing) with 3 seedlings per hole. The fertilizers were urea (N 46.4%), superphosphate (P_2_O_5_ 12.0%), and potassium chloride (K_2_O 60.0%). For each treatment, 180 kg ha^−1^ of nitrogen (N), 75 kg ha^−1^ of phosphorus (P), and 150 kg ha^−1^ of potassium (K) were applied. N fertilizer was applied at a 5:3:2 ratio of basal fertilizer/tillering fertilizer/panicle fertilizer. Among them, the basal fertilizer was applied one day before transplanting and sowing, the tillering fertilizer was applied 10 days after transplanting (at the fifth leaf stage for direct seeding), and the panicle fertilizer was applied at the third stage of young panicle differentiation. P and K fertilizers were applied one time as basal fertilizers. Each plot was separated with cement to prevent a mutual infiltration of water and fertilizer, and other management methods were carried out according to local high-yield cultivation measures.

### 4.3. Measurements and Methods

#### 4.3.1. Growth Period and Climate Data

The national meteorological monitoring station (CAWS600, China Huayun Meteorological Technology Group Co., Ltd., Beijing, China) was used to record the average daily temperature and rain fall in the rice growing season and to accurately record the main growth period of rice. The effective accumulated temperature of each growth stage was calculated using the following formula, Ke = $\sum_{i}^{n}T1-T0$, where T1 is the average daily temperature on day i and T0 is the developmental threshold temperature of the plant, which is 10 °C [39,40]. The climate data of the experimental site are shown in Figure 6.

#### 4.3.2. Yield and Yield Components

At the maturity stage, 60 holes were selected from each plot to investigate the average tillering number and then 6 representative plants were selected, and the effective panicles, spikelet per panicle, seed-setting rate, and 1000-grain weight were investigated. Finally, the actual yield was calculated and adjusted to a moisture content of 13.5%.

#### 4.3.3. Tillering Dynamics and Dry Matter Accumulation

At the jointing stage, heading stage, and maturity stage, 60 holes were selected in each plot to investigate the average tillering number. At the heading stage and maturity stage, 6 representative plants were selected according to the average tillering number, and the plants were cleaned and separated into 3 parts: stem-sheath, leaf, and panicle. All the samples were blanched at 105 °C for 30 min, transferred to 85 °C for drying to a constant weight, and weighed.

#### 4.3.4. Relative Chlorophyll Content, Net Photosynthetic Rate, and Leaf Area Index

Ten flag leaves with the same growth were selected at the beginning of the heading stage, and were then selected again 7 d, 14 d, and 21 d after the heading and maturity stage. The relative chlorophyll content was measured with a portable chlorophyll meter (SPAD-502, Konica Minolta Holdings Inc., Tokyo, Japan), and the net photosynthetic rate was measured with a photosynthetic instrument (GFS-3000, Zealquest Scientific Technology Co., Ltd., Shanghai, China). Twelve representative plants with the same growth were selected at the beginning of the heading stage, and were then selected again 7 d, 14 d, 21 d after the heading and maturity stage. The leaf area index was then calculated using the following formula: LAI = leaf length × leaf width × 0.75.

### 4.4. Statistical Analysis

Analysis of variance (ANOVA) was used to analyze the data, and means were compared based on the least significant difference (LSD) test at a 0.05 probability level using SPSS 21.0 (Statistical Product and Service Solutions Inc., Chicago, IL, USA). Figures were plotted using Origin Pro 2022 (OriginLab, Northampton, MA, USA). A structural equation model was established using SPSS Amos 28 (Statistical Product and Service Solutions Inc., Chicago, IL, USA).

## 5. Conclusions

Different crop rotation patterns and planting methods had significant effects on rice yield and the utilization of temperature and light resources. The primary climate factors influencing rice yield were identified as effective accumulated temperature and average daily temperature, exhibiting positive and negative effects on rice yield, respectively. The lower average daily temperature, coupled with an extended rice whole growth period, contributed to effective increased accumulated temperature and dry matter accumulation. As a result, grain yield under the vegetable–rice rotation pattern was 10.21–22.00% higher than the rape–rice rotation pattern and 13.99–24.49% higher than the wheat–rice rotation pattern. Both dry matter accumulation and the ratio of productive tillers under different crop rotation patterns and planting methods followed the trend of VT1 > VT2. Moreover, RT2 and WT2 exhibited a higher effective accumulated temperature than those of RT1 and WT1, resulting in increased dry matter accumulation and grain yield. In summary, the selection of appropriate planting methods for different rotation patterns facilitates optimal utilization of climate resources. The findings of this study suggested that mechanical direct seeding with the vegetable–rice rotation pattern as well as mechanical direct seeding with the rape–rice and wheat–rice rotation patterns represent optimal combinations in southwest China. However, this study has not clarified the relationship between climate and rice quality. Further research is needed to focus on the climate conditions required for the synergistic development of both yield and quality, as well as their physiological mechanisms.

## Figures and Tables

**Figure 1 plants-13-00526-f001:**
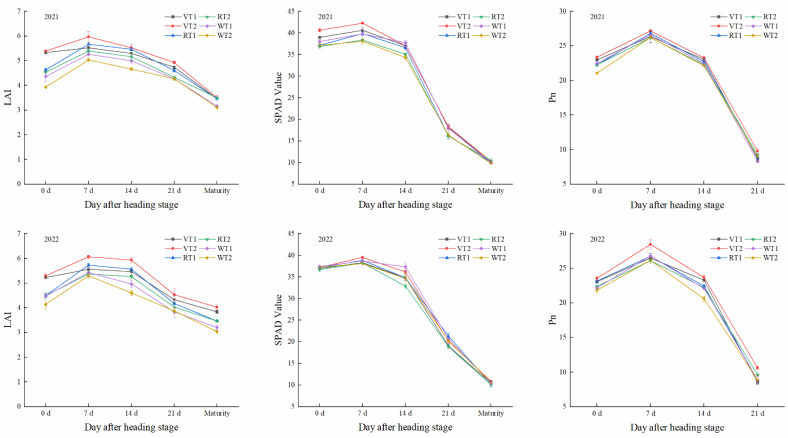
Different rotation patterns and planting methods on LAI, SPAD, and Pn of rice. V, R, and W represent vegetable–rice rotation, rape–rice rotation, and wheat–rice rotation, respectively. T1 and T2 refer to mechanical transplanting and mechanical direct seeding, respectively. LAI, SPAD and Pn refer to leaf area index, relative chlorophyll content, and net photosynthesis, respectively. C and P represent crop rotation and planting method, respectively.

**Figure 2 plants-13-00526-f002:**
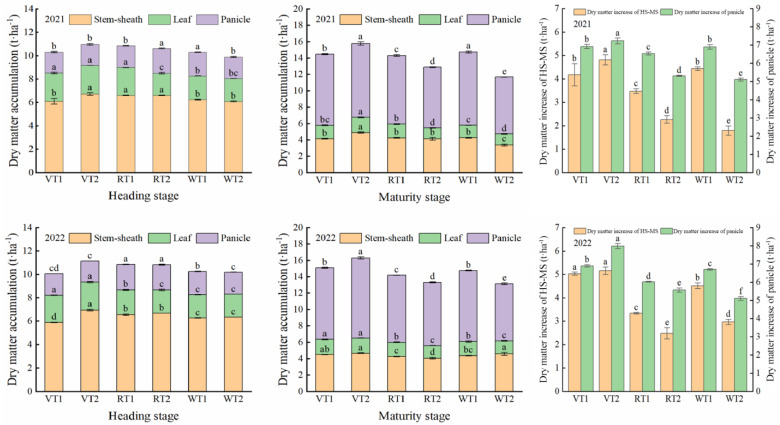
Effects of different rotation patterns and planting methods on dry matter accumulation. V, R, and W represent vegetable–rice rotation, rape–rice rotation, and wheat–rice rotation, respectively. T1 and T2 refer to mechanical transplanting and mechanical direct seeding, respectively. Values within the same organ followed by different letters are significantly different at the 0.05 probability level.

**Figure 3 plants-13-00526-f003:**
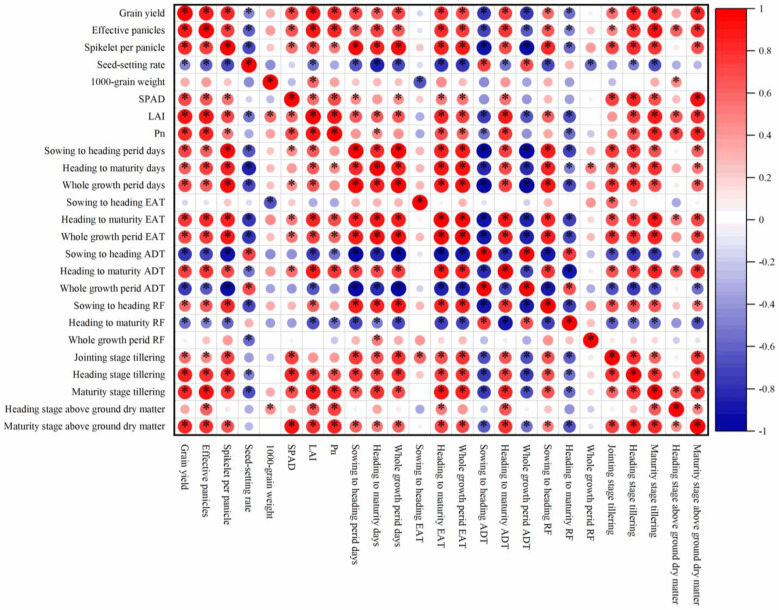
The correlation between main environmental factors and yield and its composition. LAI, SPAD, and Pn refer to leaf area index, relative chlorophyll content, and net photosynthesis, respectively. EAT, ADT, and RF refer to effective accumulated temperature, average daily temperature, and rain fall, respectively. ANOVA *p* values and symbols are defined as *, *p* < 0.05.

**Figure 4 plants-13-00526-f004:**
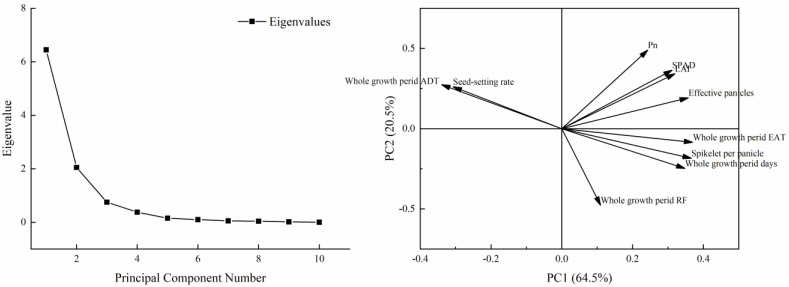
Scree plot and loading plot. PC1 and PC2 refer to principal component 1 and principal component 2, respectively.

**Figure 5 plants-13-00526-f005:**
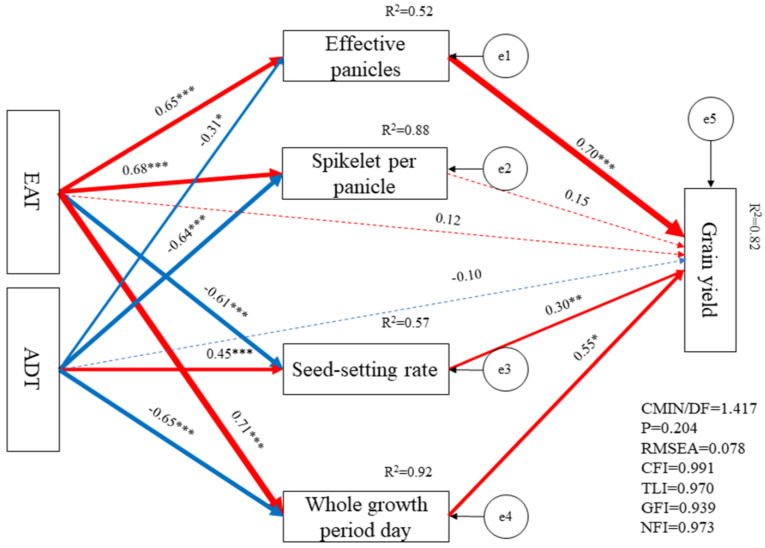
Structural equation model of yield, EAT, ADT, and yield components. EAT and ADT refer to effective accumulated temperature and average daily temperature, respectively. The arrows in the figure point to the causal relationship, the red solid line indicates a positive significant correlation, the blue solid line indicates a negative significant correlation, the red dotted line indicates a positive but not significant correlation, and the blue dotted line e indicates a negative but not significant correlation. The number refers to the effect coefficient. *, **, and *** refer to a significant correlation at a 0.05 level, 0.01 level, and 0.001 level, respectively. The R^2^ indicates the total explanatory rate of all variables pointing to the dependent variable. The e, CMIN/DF, P, RMSEA, CFI, TLI, GFI, and NFI represent the unobserved variable, ratio of chi-squared value to the degree of freedom, test *p*-value, root-mean-square error of approximation, comparative fit index, Tucker–Lewis index, goodness of fit index, and normed fit index, respectively.

**Figure 6 plants-13-00526-f006:**
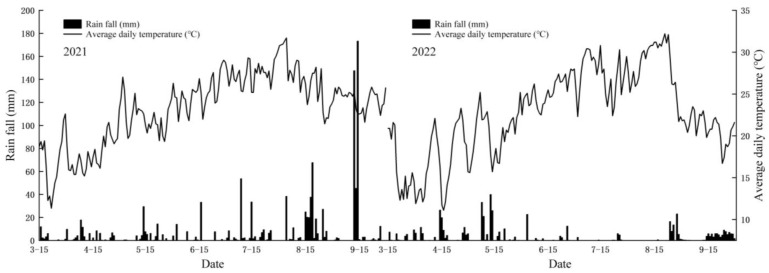
Field climate data of rice seasons in 2021 and 2022.

**Table 1 plants-13-00526-t001:** Effects of different crop rotation patterns and planting methods on the growth process of rice.

Year	Treatment	Sowing–Heading (Days)	Heading–Maturity (Days)	Whole Growth Period (Days)
2021	VT1	124 a	34 a	158 a
	VT2	103 c	30 b	133 c
	RT1	111 b	34 a	145 b
	RT2	91 e	31 b	122 e
	WT1	101 d	31 b	132 d
	WT2	85 f	29 c	114 f
2022	VT1	121 a	34 b	155 a
	VT2	100 c	34 b	134 c
	RT1	107 b	36 a	143 b
	RT2	89 e	29 c	118 e
	WT1	99 d	30 b	129 d
	WT2	81 f	30 c	111 f
F value	Y	470.32 **	3.45 ns	480.22 **
	C	7374.35 **	83.31 **	13,008.48 **
	P	18,865.16 **	169.08 **	33,235.00 **
	Y × C	0.48 ns	9.37 **	28.91 **
	Y × P	0.00 ns	3.45 ns	10.65 **
	C × P	88.55 **	39.37 **	251.96 **
	Y × C × P	12.10 **	24.58 **	36.74 **

V, R, and W represent vegetable–rice rotation, rape–rice rotation, and wheat–rice rotation, respectively. T1 and T2 refer to mechanical transplanting and mechanical direct seeding, respectively. C and P represent crop rotation and planting method, respectively. Values within the same year followed by different letters are significantly different at the 0.05 probability level. ANOVA *p* values and symbols are defined as **, *p* < 0.01; and ns, *p* > 0.05.

**Table 2 plants-13-00526-t002:** Effects of different crop rotation patterns and planting methods on main climate factors.

Year	Treatment	Sowing to Heading			Heading to Maturity			Whole Growth Period		
		EAT (°C)	ADT (°C)	RF (mm)	EAT (°C)	ADT (°C)	RF (mm)	EAT (°C)	ADT (°C)	RF (mm)
2021	VT1	1331.40 bc	20.64 e	382.80 a	581.00 a	27.26 a	262.07 de	1912.40 a	22.00 f	644.87 c
	VT2	1336.23 b	22.89 c	334.60 c	509.07 b	26.78 b	282.60 c	1845.30 c	23.77 d	617.20 d
	RT1	1353.50 a	22.12 d	369.17 b	522.40 b	26.38 c	301.43 b	1875.90 b	23.12 e	670.60 b
	RT2	1325.00 cd	24.40 b	340.60 c	456.70 c	24.89 d	256.90 e	1781.70 d	24.52 b	597.50 f
	WT1	1317.87 d	24.53 b	342.10 c	445.80 c	24.95 d	269.70 d	1763.67 d	24.24 c	611.80 e
	WT2	1335.90 b	25.53 a	384.20 a	392.03 d	23.68 e	503.97 a	1727.93 e	25.06 a	888.17 a
2022	VT1	1356.10 b	21.12 f	351.40 a	598.57 c	27.78 d	12.40 c	1954.67 b	22.56 f	363.80 a
	VT2	1317.63 d	23.09 d	265.67 c	637.73 b	28.94 a	49.73 b	1955.37 b	24.56 d	315.40 c
	RT1	1361.10 b	22.60 e	294.80 b	663.63 a	28.71 b	45.90 b	2024.73 a	24.09 e	340.70 b
	RT2	1323.90 d	24.71 b	212.60 e	532.97 d	28.59 b	68.37 a	1856.87 c	25.65 b	280.97 e
	WT1	1410.17 a	24.15 c	241.30 d	546.30 d	28.42 c	68.73 a	1956.47 b	25.13 c	310.03 d
	WT2	1345.60 c	26.41 a	65.00 f	460.50 e	25.52 e	70.40 a	1806.10 d	26.17 a	135.40 f
F value	Y	112.77 **	17.00 **	3004.23 **	520.39 **	6475.27 **	12,665.75 **	906.42 **	7120.35 **	265.19 **
	C	31.15 **	549.64 **	399.61 **	334.10 **	1781.79 **	404.49 **	290.75 **	1086.65 **	146.78 **
	P	182.70 **	620.80 **	825.78 **	249.29 **	845.68 **	383.74 **	572.80 **	1780.11 **	584.89 **
	Y × C	78.64 **	0.28 ns	460.05 **	7.29 **	305.98 **	158.15 **	23.16 **	124.15 **	1000.66 **
	Y × P	155.57 **	2.94 ns	551.22 **	0.36 ns	63.34 **	115.07 **	31.41 **	89.64 **	1069.86 **
	C × P	6.71 **	4.92 *	3.12 ns	37.96 **	581.80 **	271.96 **	63.15 **	662.00 **	2288.09 **
	Y × C × P	35.22 **	9.79 **	172.90 **	48.20 **	323.12 **	401.42 **	59.46 **	1.57	1023.60 **

V, R, and W represent vegetable–rice rotation, rape–rice rotation, and wheat–rice rotation, respectively. T1 and T2 refer to mechanical transplanting and mechanical direct seeding, respectively. EAT, ADT, and RF refer to effective accumulated temperature, average daily temperature, and rain fall, respectively. C and P represent crop rotation and planting method, respectively. Values within the same year followed by different letters are significantly different at the 0.05 probability level.A NOVA *p* values and symbols are defined as *, *p* < 0.05; **, *p* < 0.01; and ns, *p* > 0.05.

**Table 3 plants-13-00526-t003:** Effects of different rotation patterns and planting methods on tillering dynamics of rice (×10^4^ ha^−1^).

Year	Treatment	Jointing Stage	Heading Stage	Maturity Stage	Ratio of Productive Tillers (%)
2021	VT1	407.22 ± 10.84 b	300.56 ± 15.12 a	271.67 ± 3.89 c	66.75 ± 2.40 d
	VT2	332.22 ± 6.94 c	300.00 ± 5.77 a	291.48 ± 4.63 a	87.74 ± 0.44 a
	RT1	400.56 ± 3.47 b	288.33 ± 13.02 b	281.11 ± 4.81 b	70.18 ± 1.18 d
	RT2	307.78 ± 1.92 d	257.78 ± 5.85 c	247.22 ± 4.19 de	80.32 ± 0.92 b
	WT1	421.67 ± 6.01 a	283.52 ± 4.73 b	254.17 ± 3.00 d	60.28 ± 1.14 e
	WT2	313.89 ± 2.55 d	257.41 ± 7.06 c	238.89 ± 10.18 e	76.12 ± 3.61 c
2022	VT1	409.44 ± 1.92 b	297.78 ± 1.92 ab	273.89 ± 2.55 b	66.90 ± 0.88 d
	VT2	332.22 ± 6.94 c	301.11 ± 10.72 a	290.00 ± 3.33 a	87.31 ± 1.31 a
	RT1	402.78 ± 1.92 b	288.89 ± 2.55 b	282.78 ± 2.55 a	70.21 ± 0.44 d
	RT2	308.89 ± 8.39 d	252.22 ± 1.92 c	245.56 ± 7.70 c	79.57 ± 4.40 b
	WT1	423.33 ± 4.41 a	289.44 ± 4.19 b	253.33 ± 1.67 c	59.84 ± 0.80 e
	WT2	314.44 ± 5.09 d	258.89 ± 5.09 c	234.44 ± 5.09 d	74.59 ± 2.77 c
F value	Y	0.44 ns	0.00 ns	0.20 ns	0.54 ns
	C	24.21 **	67.53 **	154.90 **	71.42 **
	P	2251.36 **	80.08 **	46.44 **	503.98 **
	Y × C	0.01 ns	0.67 ns	0.31 ns	0.14 ns
	Y × P	0.14 ns	0.24 ns	1.10 ns	0.36 ns
	C × P	22.76 **	23.33 **	85.62 **	21.66 **
	Y × C × P	0.01 ns	0.47 ns	0.00 ns	0.01 ns

V, R, and W represent vegetable–rice rotation, rape–rice rotation, and wheat–rice rotation, respectively. T1 and T2 refer to mechanical transplanting and mechanical direct seeding, respectively. C and P represent crop rotation and planting method, respectively. Values within the same year followed by different letters are significantly different at the 0.05 probability level. ANOVA *p* values and symbols are defined as **, *p* < 0.01; and ns, *p* > 0.05.

**Table 4 plants-13-00526-t004:** Effects of different rotation patterns and planting methods on rice yield and yield components.

Year	Treatment	Effective Panicles (×10^4^ ha^−1^)	Spikelet per Panicle	Seed-Setting Rate (%)	1000-Grain Weight (g)	Grain Yield (t ha^−1^)
2021	VT1	269.72 ± 1.94 b	177.92 ± 1.70 a	81.81 ± 0.00 c	23.13 ± 0.07 b	9.60 ± 0.08 b
	VT2	287.78 ± 1.92 a	169.51 ± 0.93 b	82.62 ± 0.01 bc	23.10 ± 0.02 b	9.92 ± 0.02 a
	RT1	270.56 ± 14.17 b	171.00 ± 1.33 b	80.73 ± 0.01 d	23.01 ± 0.04 b	8.32 ± 0.10 c
	RT2	251.11 ± 10.72 c	159.45 ± 3.29 cd	83.33 ± 0.01 ab	23.33 ± 0.03 a	7.68 ± 0.14 e
	WT1	249.44 ± 4.81 c	163.25 ± 0.81 c	83.67 ± 0.01 a	22.36 ± 0.03 d	8.02 ± 0.09 d
	WT2	245.56 ± 1.92 c	158.35 ± 4.25 d	83.33 ± 0.01 ab	22.63 ± 0.12 c	7.67 ± 0.07 e
2022	VT1	273.89 ± 2.55 c	178.72 ± 1.12 a	81.06 ± 0.01 cd	23.26 ± 0.05 ab	8.92 ± 0.09 a
	VT2	290.00 ± 3.33 a	169.60 ± 5.09 b	82.08 ± 0.01 bc	23.14 ± 0.08 ab	9.00 ± 0.37 a
	RT1	279.63 ± 1.60 b	171.27 ± 0.94 b	80.22 ± 0.01 d	23.08 ± 0.11 bc	8.45 ± 0.12 b
	RT2	246.67 ± 3.33 e	152.66 ± 5.07 d	83.40 ± 0.01 ab	23.34 ± 0.31 a	7.81 ± 0.04 d
	WT1	253.33 ± 1.67 d	163.10 ± 1.77 c	84.15 ± 0.01 a	22.35 ± 0.03 d	8.09 ± 0.04 c
	WT2	245.56 ± 1.92 e	155.67 ± 2.22 d	83.38 ± 0.01 ab	22.84 ± 0.01 c	7.63 ± 0.07 d
F value	Y	1.77 ns	2.20 ns	0.96 ns	4.28 ns	22.01 **
	C	97.59 **	76.13 **	16.96 **	129.29 **	399.26 **
	P	7.11 *	110.33 **	13.88 **	28.75 **	36.50 **
	Y × C	0.04 ns	1.25 ns	0.73 ns	0.30 ns	38.72 **
	Y × P	2.98 ns	3.25 ns	0.35 ns	0.12 ns	1.49 ns
	C × P	44.71 **	7.71 **	13.19 **	14.72 **	28.14 **
	Y × C × P	0.92 ns	0.98 ns	0.35 ns	1.83 ns	0.51

V, R, and W represent vegetable–rice rotation, rape–rice rotation, and wheat–rice rotation, respectively. T1 and T2 refer to mechanical transplanting and mechanical direct seeding, respectively. C and P represent crop rotation and planting method, respectively. Values within the same organ followed by different letters are significantly different at the 0.05 probability level. ANOVA *p* values and symbols are defined as *, *p* < 0.05; **, *p* < 0.01; and ns, *p* > 0.05.

**Table 5 plants-13-00526-t005:** The scores and rankings under different rotation patterns and planting methods.

Treatment	PC1 Score	Ranking	PC2 Score	Ranking	Comprehensive Scores	Ranking
VT1	2.54	1	−1.55	6	1.32	2
VT2	2.41	2	1.74	1	1.91	1
RT1	2.18	3	−0.55	5	1.30	3
RT2	−2.65	5	−0.11	4	−1.73	5
WT1	−1.21	4	0.16	3	−0.75	4
WT2	−3.28	6	0.39	2	−2.03	6

V, R, and W represent vegetable–rice rotation, rape–rice rotation, and wheat–rice rotation, respectively. T1 and T2 refer to mechanical transplanting and mechanical direct seeding, respectively. PC1 and PC2 refer to principal component 1 and principal component 2, respectively.

**Table 6 plants-13-00526-t006:** Soil properties of the topsoil layer (0–20 cm) at the experimental sites.

Year	Preceding Crop	Total N (g kg^−1^)	Total P (mg kg^−1^)	Organic Matter (g kg^−1^)	Available N (mg kg^−1^)	Available P (mg kg^−1^)	Available K (mg kg^−1^)	pH
2021	Vegetable	1.60	325.12	29.12	118.46	9.11	130.44	6.04
	Rape	1.64	427.43	31.35	114.40	10.83	140.48	6.26
	Wheat	1.64	398.95	30.19	112.39	10.35	157.08	6.41
2022	Vegetable	1.59	333.65	30.32	116.45	9.37	131.55	5.93
	Rape	1.65	427.45	30.71	113.49	10.73	140.07	6.31
	Wheat	1.61	400.56	31.88	112.54	10.79	154.31	6.41

## Data Availability

Data are contained within the article.

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
