# Peer review of "Responses of Yield and Photosynthetic Characteristics of Rice to Climate Resources under Different Crop Rotation Patterns and Planting Methods"

_plants, 2024, doi:10.3390/plants13040526_

Round 1

Reviewer 1 Report

Comments and Suggestions for Authors

The paper „Response of yield and photosynthetic characteristics to climate resources under different crop rotation patterns and planting methods” is current.

The authors highlighted  the characteristics of rice growth, development and yield formation and their relationships with climate factors under three rotation patterns -vegetable-rice, rape-rice, and wheat-rice and two planting methods - mechanical transplanting and mechanical direct seeding.

I propose the publish of the paper after they will expanding the conclusions. In the Conclusion, briefly add  perspectives in the continuation of research and trials.

Reviewer 2 Report

Comments and Suggestions for Authors

An interesting study relating various crop rotations and rice planting methods to climate factors that determine biomass formation and yield. However the following queries need to be attended to.

Putting the Materials and Methods section after Discussion, rather than after Introduction as is normally done, I find disconcerting. Logically, one cannot assess the results until after having read the details of how the study was done.

Manuscripts for review should have line numbers.

Although mostly clearly written the manuscript would benefit from further minor English language editing; e.g. last sentence second last para of Introduction: “There is still limited study can be found in ---” can be better expressed as “There is limited study ---”.

Title

Include “of rice” after photosynthetic characteristics.

Introduction

Even though abbreviations are given in the Abstract, from the Introduction onwards they should be given again when first mentioned in the text.

Materials and Methods

Suggest this section be included after Introduction section.

Table 5. What are the units for available N, P and K?

Was the rice rainfed or irrigated, and if irrigated give details?

Were vegetables, rape and wheat grown in the same plots in between the2021 and 2022 rice crops?

Fig. 6. Are the temperatures plotted average daily temperatures, and how are they calculated?

4.3.2. What area of plots was harvested for yield determination? Were any borders left?

4.3.3. What is “jointing stage”?

Results

Table 1. For 2021 VT1 the total growth period is not the sum of the two separate growth period, as it is for all other treatments. Why?

Table 2. For heading to maturity the rainfall appears to be five times greater in 2021 as compared to 2022 – is this correct? What are the units for rainfall?

There is no discussion of Fig 3 in the Results text. And there is no labelling on the x axis if Fig. 3.

Fig. 5. Meaning and relevance of last sentence of caption unclear.

Discussion

3.1. The results are interpreted as caused by various climate factors which varied according to sowing date. Can this be interpreted just as analogous to a date of sowing trial or could there by any residual effects of the preceding crop on rice, e.g. residual fertilizer differences, disease, crop residue, etc.?

Comments on the Quality of English Language

Mostly excellent quality but some expression could be improved - needs minor language editing. Excessive use of "respectively". Better to express as "Treatment A had a value of x and Treatment B a value of y". Rather than "Values for Treatments A and B were x and y, respectively".
